

# Constraining the depth of the lithosphere-asthenosphere boundary in tectonically complex regions using locally adjusted lithological forward models and seismic velocities

Nóra Liptai[1], Dániel Kalmár[2], István János Kovács[1]

[1] HUN-REN Institute of Earth Physics and Space Science, Sopron, 9400, Hungary
[2] Kövesligethy Radó Seismological Observatory, HUN-REN Institute of Earth Physics and Space Science, Budapest, 1112, Hungary
*Correspondence to*: Dániel Kalmár (kalmar.daniel@epss.hun-ren.hu)

**Abstract.** In this study, we employ 1D migration of S-to-P receiver functions to determine the lithosphere-asthenosphere boundary (LAB) depth beneath 41 seismological stations in the Pannonian Basin (Hungary). Our approach incorporates forward models of seismic velocity profiles tailored to local lithospheric structures, allowing for an improved constraint on LAB depths in comparison with global velocity models.

We systematically evaluate the impact of crustal structure, mantle composition, temperature variations, and partial melt on seismic velocities. Global velocity models, while effective for large-scale studies, have significant limitations in resolving lithospheric-scale structures due to their coarse parameterization. Our forward models incorporate variations in sedimentary layer velocities, Moho depth, and LAB-associated velocity reductions, leading to LAB depth estimates that differ from those derived using the IASP91 model. Notably, 1 vol.% partial melt below the LAB results in a pronounced velocity decrease, whereas metasomatism and volatile-bearing phases may cause localised velocity reductions but have limited effect on the determination of LAB depths.

LAB depths obtained using local versus global velocity models reveal a strong correlation, yet notable discrepancies exist at stations with complex lithospheric structures or thick sedimentary cover, as further supported by anomalies in surface heat flow data. Our results demonstrate that locally calibrated seismic velocity models offer a more accurate representation of the lithosphere-asthenosphere transition than global models, particularly in regions with complex tectonic and thermal histories. Integrating geophysical and petrological approaches is key when investigating lithospheric structure, as well as the combined interpretation of different factors that shape seismic LAB signatures.

## 1 Introduction

The lithosphere-asthenosphere boundary (LAB) is commonly understood as the horizon within the Earth's upper mantle,
where a change in viscosity and other physical and chemical properties separates the outermost rigid layers from the rheologically weaker asthenosphere, and thus facilitates plate tectonics. However, the change of these physical properties with depth is not synchronous, therefore, the exact definition of the LAB is elusive and there can be large variations





regarding its depth (Rychert et al., 2020 and references therein). The 'thermal' LAB is considered to be the 1300°C isotherm, which is located within the zone where heat transport gradually changes from conductive to convective (e.g., Artemieva,

2009). It has been shown that geophysical parameters such as electrical resistivity and seismic wave velocity start to drop at lower temperatures (Pollack and Chapman, 1977), marking the top of the low velocity zone. While several explanations have been proposed to account for this resistivity and velocity drop, including change in the deformation mechanism of olivine, anisotropy, grain size or modal composition, these are all subsolidus processes. Recently, it is accepted that small amount of partial melt represents the key role in forming the low velocity zone (e.g., Anderson and Spetzler, 1970; Rychert et al., 2007;

Chantel et al., 2016; Selway and O'Donnell, 2019). Kovács et al. (2021) proposed that small P-T changes at the stability limit of pargasitic amphibole (~1100°C or ~3 GPa; Green et al., 2010) result in continuous melting and pargasite crystallisation, which could be responsible for the seismic discontinuity. In areas with thin lithosphere and high heat flow (i.e., oceanic and extensional continental plates), this corresponds to the LAB, whereas in areas with thicker lithosphere and low heat flow (i.e., cratons), to the globally observed mid-lithospheric discontinuity (MLD).

Receiver function analysis is a widely applied tool in determining the depth of seismic discontinuities, because seismic waves convert from shear to compressional (S-to-P) or vice versa (P-to-S) upon passing through a discontinuity separating materials with different physical properties. Its widespread application is particularly advantageous in regions with dense station coverage, as it facilitates the creation of a coherent lithospheric thickness image for the area (e.g., Kind et al., 2020). Unlike the sharp Moho interface, the LAB lacks a distinct seismological discontinuity, making its accurate determination

challenging. However, recent seismological data, coupled with the S-to-P receiver function technique, have significantly improved the precision of LAB determination (Hu et al., 2011). The methodology relies on S-to-P wave conversions, derived by deconvoluting S and P components from three-component, broadband seismograms following rotation (Kind et al., 2012). Most commonly, receiver function studies use global velocity 1D models such as the AK135 (Kennett et al., 1995) or the IASP91 (Kennett and Engdahl, 1991) models to determine the depth of the LAB (or sensu stricto, the top of the low velocity

zone). However, global models exclude the possibility of considering local characteristics that could affect the 1D velocity profile, such as thickness of the crust and lithosphere, and the velocity drop expected at the LAB. Therefore, when one aims to determine LAB depth using receiver functions, the question arises how accurate these global velocity models are, especially if the study area has a unique tectonic environment.

To test this, we constructed 1D velocity models in the Pannonian Basin (Central Europe, inset in Fig.1), taking local

properties into account, such as heat flow, sediment thickness, crustal structure (including Conrad and Moho) and presumed LAB-depth. The Pannonian Basin is an ideal candidate for this test, as it has anomalously thin lithosphere (Babuška and Plomerová, 1988; 1992; 1993; Horváth, 1993; Tari et al., 1999; Kalmár et al., 2023), it is well-covered by permanent and temporary seismic stations, sediment and crustal thickness maps are available (e.g., Balázs et al., 2016; Horváth et al., 2015; Kalmár et al., 2021), and numerous xenolith data offer information on mantle composition (e.g., Liptai et al., 2017; Aradi et

al., 2020; Patkó et al., 2022). We use these 'custom' velocity models for migration in S-to-P receiver function analyses to determine LAB depths, compare them with those resulting from using global velocity models, as well as with previous LAB



maps in the area. Furthermore, in our models we also consider the effect of small amount of partial melt on seismic velocities. As a result, this study offers new insights and a more accurate depiction of the geology and geochemistry of the research area by implementation of the 1D migration of the receiver function with enhanced velocity models tailored to the specific lithospheric configuration.

## 2. LAB depths in the Pannonian Basin

The Pannonian Basin, surrounded by the Alps, Carpathians and Dinarides (Fig. 1), is part of a Neogene back-arc basin system. It is composed of two microplates, AlCaPa and Tisza(-Dacia), which were juxtaposed during the late Oligocene, then suffered major extension-driven lithospheric thinning in the Miocene (Csontos et al., 1992; Horváth, 1993). From late Miocene to recent times, large-scale tectonic inversion dominates the region, attributed to the convergence and rotation of the Adria block towards the European plate (Horváth and Cloetingh, 1996; Bada et al., 2007; Porkoláb et al., 2023).

In the past few decades, several studies have presented LAB depth maps on the Pannonian Basin using different geophysical approaches. Babuška and Plomerová (1993) determined lithospheric thicknesses in Central Europe based on seismic tomography, heat flow and magnetotelluric data. Based on the same data, Horváth (1993) prepared a slightly modified map focusing on the Carpathian-Pannonian region, and this map was adapted and further improved by Tari et al. (1999) and Horváth et al. (2006). According to this map, lithospheric thickness in the central regions of the Pannonian Basin is ~60 km or even lower. 3D gravity modelling by Tasarova et al. (2009) showed similar results, with LAB depths intensively increasing towards the northern and western parts of the basin. The lithospheric thickness map compiled by Bielik et al. (2010), combining 2D gravity modelling and heat flow data, proposed ~80 km LAB depths even for the centre of the Pannonian Basin. However, all of these maps have relatively low resolution and high uncertainty, which stands in contrast with the high number of seismological stations currently operating in Hungary and surrounding countries.

Most recently, using 1D migration of the stacked S-to-P receiver functions, Kalmár et al. (2023) published a negative phase depth map for the broader Carpathian-Pannonian region. This negative phase depth can be considered as the LAB in areas with thin lithosphere, such as the Pannonian Basin, or, in areas with thicker lithosphere, a mid-lithospheric discontinuity. While this map has a better resolution due to the coverage of seismic stations, the velocity model used for the migration was the IASP91 global velocity model (Kennett and Engdahl, 1991). Although Kalmár et al. (2023) has taken sediment thickness into account, the IASP91 global model still assumes a uniformly thick crust (~35 km), and does not consider the decrease of seismic velocity beneath the LAB. Furthermore, neither the effect of melts and fluids in the upper mantle nor realistic upper mantle compositions are considered. In our present study we shed light on these factors.





**Figure 1. Location of seismological stations used in this study within the Carpathian-Pannonian region in Central-Eastern Europe (inset). White dashed lines indicate lithospheric thicknesses in km after the map of Tari et al. (1999). Coloured triangles represent different networks of seismic stations. Nowadays these stations operate as part of the AdriaArray Seismic Network. Abbreviations of topographic units: EA – Eastern Alps, WC – Western Carpathians, EC – Eastern Carpathians, SC – Southern Carpathians, PB – Pannonian Basin, DI – Dinarides.**

## 3. Methods

Seismic forward models were constructed for 41 seismological stations in Hungary (Fig. 1), to use them as velocity models for 1D migration and LAB depth constraining. The stations include permanent stations of the Hungarian National Seismological Network (Gráczer et al., 2018), and temporary stations part of the AlpArray (Hetényi et al., 2018) and PACASE (Pannonian-Carpathian-Alpine Seismic Experiment; (Schlömer et al., 2024 projects which are operating since 2016 and 2019, respectively, currently within the framework of the AdriaArray initiative. The velocity profiles, along with the applied parameters (heat flow, sediment and crustal thickness, LAB depth determined using the IASP91 global model) is provided in Table S1.



To create 1D seismic forward models, the first step is to build theoretical lithology columns for each seismic station. For
this, the depth of horizons separating major lithological changes need to be known, such as the sediment thickness, Conrad,
Moho and LAB depths as well. Sediment thickness, Conrad and Moho depth maps are available for Hungary; for our models
we used the most recent data of Kalmár et al. (2021) based on P-to-S receiver function analysis. Even though the aim of 1D
migration is to find the depth of the LAB, the forward model needs an initially assumed LAB to predict the distribution of
seismic velocities by depth. Therefore, we used the negative phase depths of Kalmár et al. (2023) for initial 'model' LAB
depths beneath the Pannonian Basin.

For the 1D velocity models, the Excel workbook of Abers and Hacker (2016) was used, which contains physical properties
data for many crustal and mantle mineral endmembers, and allows the calculations of P- and S-wave velocities of crustal and
mantle rocks with known mineral composition, pressure and temperature. Composition of the mantle was averaged from a
large number of mantle xenoliths described from the Carpathian-Pannonian region (Liptai et al., 2017; Aradi et al., 2020;
Patkó et al., 2022), resulting in an average lherzolitic composition. As significant compositional changes are not expected in
the mantle of the region, it was used for the models of all stations. The effect of different compositions (i.e., changes in the
proportion of olivine and pyroxenes) on seismic velocity is examined within the parameter tests (see section 4 below).
Similarly to the mantle, crust composition was uniformised for the models; we used a composition that starts as granitic
gneiss in the uppermost crust and gradually changes to mafic granulite and garnet granulite (Christensen and Mooney, 1995).
This compositional model was applied in proportion to the different crustal thicknesses of the stations. Although crustal
structures are considered more complex, this uniform composition represents a sufficiently realistic approximation, and
calculated velocities are in good agreement with the results of Kalmár et al. (2021).

Temperatures at depth were estimated using geotherms attributed to heat flow values between 120 and 50 mW/m$^2$, assuming
1300°C at 120 km depth (Kovács et al., 2017). Heat flow values were determined for each station based on the map of
Lenkey et al. (2021). During the construction of the velocity forward models, sediments were also taken into account where
present; velocity values of the top and bottom of the sediment layer were adopted from Kalmár et al. (2021). The resulting
velocity profiles reflect a 'dry' lithology column without the presence of melt. To include the effect of small amount of
partial melt below the LAB, we used the equations of Chantel et al. (2016), who described the empirical correlation between
melt proportion and seismic velocities by experimental work. The effect of different amounts of melt are detailed in the
parameter test (see Section 4). The velocity models discussed in this paper are presented in Table S2.

Raw S-to-P receiver functions were computed using the iterative time-domain deconvolution method (Ligorría & Ammon,
1999) with 300 iterations (Kalmár et al., 2023). The resulting S-to-P receiver functions (Figure S3) were then converted to a
depth range with the S-to-P conversion negative peak using the 1D migration method (Kalmár et al., 2023), a time-to-depth
conversion based on a 1D velocity model beneath the seismic station. For the migration, the velocity models were used in a 5
km resolution with the bottom at 120 km depth.



## 4. Parameter tests

Seismic wave velocity may vary depending on temperature, pressure, mineral composition, and melt presence. Parameter tests were carried out (Table S2) to determine the extent of these effects, and to constrain the uncertainty of the velocity columns resulting from not knowing precise mantle values. We chose one station, PSZ, because of its good quality receiver function data, to test the parameters on a wide scale, and to pick narrower ranges for building models for the other stations.

### 4.1. Heat flow

While pressure can be reliably calculated for a given depth, the rate of temperature increase (geotherm) is more challenging as it may vary regionally, depending on a number of factors, most commonly lithospheric architecture and history. Heat flow is a generally accepted parameter to conclude the geotherm of a specific area (Artemieva, 2009 and references therein). In Hungary, heat flow varies on a wide range, from 90-110 mW/m$^2$ on the eastern and southern regions (Great Hungarian Plain), to as low as 30-50 mW/m$^2$ on the central-western part (Transdanubian Range) (Lenkey et al., 2021). The high heat flow values are linked to the extensional lithospheric thinning, whereas the low ones are associated with the cooling effect of karst systems.

We calculated velocity profiles for geotherms based on heat flow values between 50 and 120 mW/m$^2$ (Kovács et al., 2017). Since the forward models were made for each station using the corresponding heat flow based on the heat flow map of Lenkey et al. (2021), the aim was to see the error of using inaccurate geotherm. Fig. 2 shows that the major difference occurs in the lower crust and uppermost mantle (between ~20-70 km depth), where P-wave velocity difference is ~0.15 km/s between the 120 and 70 mW/m$^2$ heat flow profile. The 50 mW/m$^2$ heat flow profile shows stronger discrepancy with a maximum of 0.3 km/s (3.8 %) difference compared to the 120 mW/m$^2$ profile. However, this much error is not expected in the forward models; the uncertainty of heat flow and consequently the geotherm applied for a station is not expected to be higher than 10-20 mW/m$^2$, which would result in minimal difference. Therefore, we can conclude that the overall effect of temperature is relatively modest on seismic velocities.

### 4.2. Mantle mineral composition

According to the database of Abers and Hacker (2016), the major silicate constituents of the mantle vary in a narrow range. Forsterite produces the highest velocities, whereas enstatite and diopside, the dominant pyroxene endmembers are lower by ~0.8 km/s (P-wave) and ~0.5 km/s (S-wave). For comparison, fayalitic olivine would produce a more drastic velocity drop (Fig. 3). If no other parameter (e.g., melt content) is considered, the velocities show only minimal decrease with depth (Fig. 3). We calculated velocities for realistic mantle compositions of lherzolite, harzburgite and dunite, which resulted in respectively 7.75, 7.82 and 7.96 km/s for P-waves and 4.40, 4.44 and 4.49 km/s for S-waves at 60 km depth. This is a 2.5% (P) and 2.0% (S) difference between dunitic and lherzolitic lithologies. Although there may be other lithologies (e.g., pyroxenites) present in the mantle under the stations in this study, their proportion is likely negligible. The modal



compositions of large number of xenoliths in the Carpathian-Pannonian region suggests that lherzolites can be considered dominant. Therefore it is safe to use a uniform lherzolitic composition for the forward models, as even occasional harzburgitic or dunitic segments would produce only minimal difference.

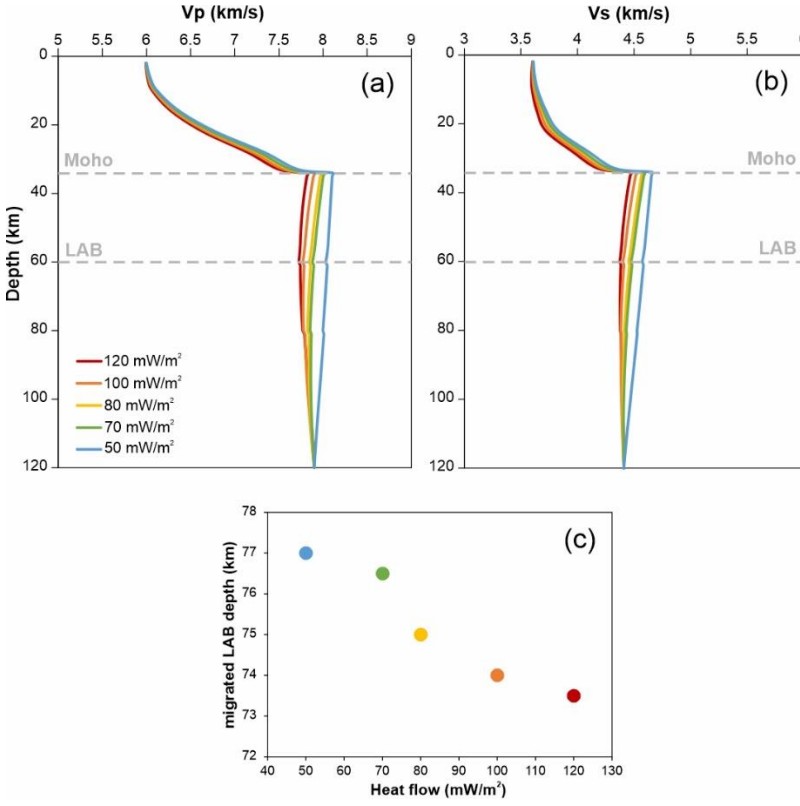

**Figure 2. a, b: Velocity profiles for P- and S-wave calculated using geotherms based on different surface heat flows (Kovács et al., 2017) for PSZ station. c: LAB depths resulting from 1D migration using velocity models based on the different surface heat flow depicted on a) and b). The models were calculated with a lherzolitic mantle composition, no pargasite accumulation above the LAB and no melt in the asthenosphere.**

## 4.3. Pargasite

As proposed by previous studies, pargasitic amphibole may have an effect on forming the LAB or mid-lithospheric discontinuities (Green et al., 2010; Selway et al., 2015; Kovács et al., 2021). According to the 'pargasosphere hypotheses' proposed by Kovács et al. (2021), hydrous melts or fluids crossing the horizon of pargasite stability, either by rising or due to minute P/T changes, could lead to pargasite enrichment in this mantle layer. Therefore, we have included in the parameter tests the seismic effect of potentially large amounts (up to 10 vol.%) of pargasite accumulated directly above the LAB, exponentially decreasing in the span of 5 km.

As shown on Fig. 4, the maximum of 10 vol%. pargasite causes a 0.08 and 0.06 km/s drop in the P- and S-velocities, respectively, which accounts to a 1% difference. Although larger proportions of amphibole have also been proposed to





explain mid-lithospheric discontinuities, this is not supported by direct evidence, such as amphibole content of xenoliths
(Selway et al., 2015). Therefore, a pargasite-rich layer by itself is not likely to have a major effect on the seismic LAB.

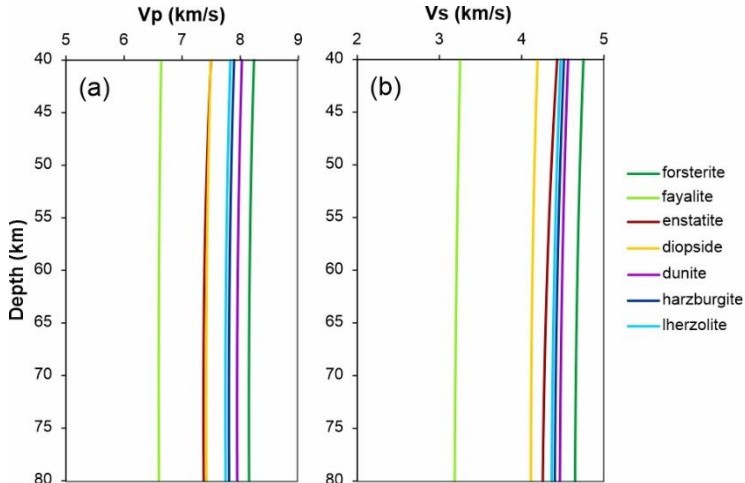

**Figure 3. a, b: P- and S-wave velocity models for the uppermost mantle with different minerals and common mantle rocks (lherzolite: 70% olivine, 20% orthopyroxene, 10% clinopyroxene; harzburgite: 80% olivine, 18% orthopyroxene, 2% clinopyroxene; dunite: 100% olivine with Fo90)**

### 4.4. Partial melt

We tested the effect of small amount of partial melt on seismic velocities within the range of melt proportions used in the experimental work of Chantel et al. (2016). In their work, they proposed that melt only affects seismic velocities if it can form a continuous network on the grain boundaries. This could not be achieved with 0.1 vol%, however, 0.5 vol% proved sufficient. Compared to the temperature and modal composition, melt content causes a more significant drop in seismic

velocities (Fig. 5). The differences in both Vp and Vs compared to melt-free mantle range from 0.3 to 1.2 km/s, which in percentages are 3-14 % for Vp and 6-27 % for Vs for the melt proportions of 0.5 – 4 vol% (see exact values on Fig. 5). As the seismic velocity reduction experienced at the LAB and mid-lithospheric discontinuities falls around 3-10 % (Rychert and Shearer, 2009 and references therein), melt proportions above 1.5 vol% do not seem realistic. Faul (2001) also argued that melt content above 1 vol% may not be retained in the upper mantle but become mobile. In contrast, even 0.5 vol% of melt

produces a lot stronger velocity drop than large amounts of accumulated pargasite above the low velocity zone.

## 5. Results

### 5.1. Model set on 4 stations

It is evident that melt content has the most significant effect on seismic velocities, and differences in modal composition and temperature (heat flow) are not negligible either. Heat flow can be used locally for each station based on the map of Lenkey

et al., 2021, and modal composition is not expected to have major variations; however, the amount of partial melt is not



known. Furthermore, the building of velocity profiles requires a presumed LAB depth. Therefore, as a next step, we created a set of forward models for four selected stations (PSZ, EGYH, MORH and LTVH; Fig. 1), to test how the variation of melt content and different initial LAB depths influence the 1D migration. The stations were selected to represent different regions of Hungary in terms of heat flow, sediment coverage, crustal thickness, and previously determined LAB depth, as well as the

AlCaPa and Tisza-Dacia microplates.

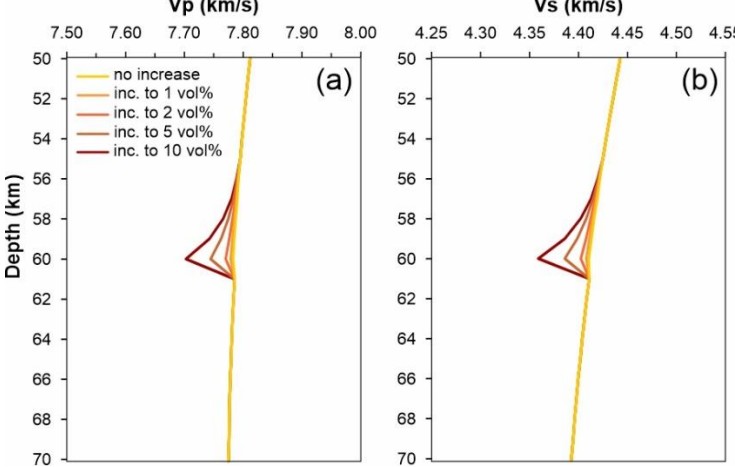

**Figure 4.** Effect of pargasite accumulation above the LAB on the calculated P- (a) and S- (b) wave velocity models.

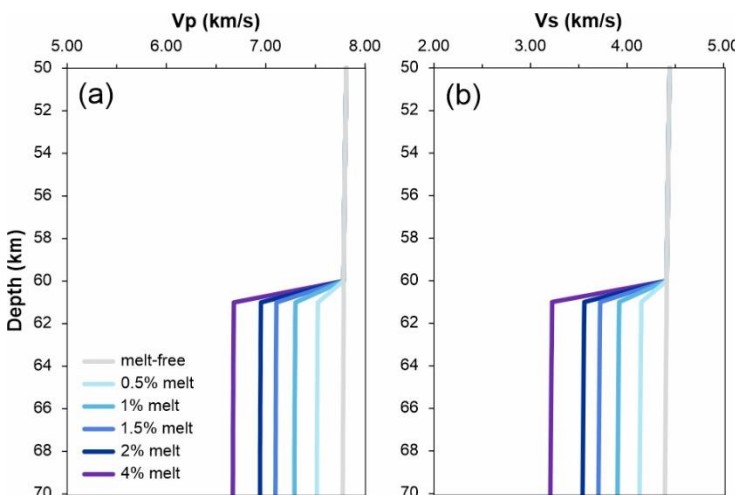

**Figure 5.** Effect of different amount of partial melt on the P- (a) and S- (b) wave velocities at the LAB.









**Figure 6.** Comparison of the presumed LAB depth and melt proportion with the 1D migrated LAB depth using different velocity models for four selected stations: PSZ (a, b), EGYH (c, d), MORH (e, f), LTVH (g, h).

Station PSZ is located in NE Hungary, on the Alcapa microplate, upon Miocene andesitic bodies and negligible amount of sediment. The other station on Alcapa is EGYH on the western part of the country (Fig. 1), in the Little Hungarian Plain in

the vicinity of the Transdanubian Range; sediment thickness is moderate (~2 km) and the LAB has been estimated at around 80 km depth as it deepens toward the Alps (Babuška and Plomerová, 1993; Horváth, 1993). On the Tisza microplate, station LTVH in the east represents the part of the basin most affected by extension with thick sediment coverage (> 4 km) and very shallow LAB (45-60 km; Kalmár et al., 2023 and Tari et al., 1999, respectively). Station MORH is located on the western part of the Tisza unit, on Paleosoic granite and no sediment coverage; negative phase depths of Kalmár et al. (2023)

indicated LAB depths at 75 km, and an anomalously high surface heat flow (120 mW/m$^2$) has been described in this area by Lenkey et al. (2021). These stations are thought to cover a wide enough range of tectonic regimes in the Carpathian-Pannonian region.

The models vary melt proportions (0 – 4 vol%) and LAB depths with the initial value taken from the negative phase depth data of Kalmár et al. (2023), supplemented with +- 5 and 10 km depth variations which were adjusted based on the results

(see below).  We compared the distribution of migrated LAB depths with respect to the amount of partial melt and the presumed LAB (Fig. 6).

Generally, increasing melt content pushes the migrated LAB towards shallower depths. At greater presumed LAB depths, however, the melt has no or only minimal effect, whereas at shallower presumed LAB, the melt effect appears more pronounced. If we compare presumed vs. migrated LAB depth (Fig. 6. b, d, f, h), it becomes clear that this melt effect has an

apparent maximum and minimum. The maximum is reached at a certain presumed LAB depth, above which the migrated depths remain constant (observable best in the case of EGYH and MORH), due to the depth of the velocity change falling beyond the negative phase on the receiver function. Furthermore, at shallower presumed LAB depths, melt contents, even higher proportions, do not significantly change the migration results, e.g., in case of MORH (Fig. 6 f), migrated depths stay quasi constant at presumed LABs shallower than 60 km. It is noteworthy that none of the melt content curves produce a

straight correlation between presumed and migrated LAB depths.

## 5.2. Models of rest of the stations

We have extended the same model sets to the permanent stations (Fig. 1), excluding the 2 and 4 vol.% melt content, as these are likely less realistic. The results generally support those of the four selected stations described above. A 'maximum' migrated LAB (i.e., at which greater presumed LAB depth does not produce greater migrated LAB at any melt content) is

sometimes reached at the same depth where it was determined using the IASP91 global velocity model, but usually at 5 or 10 km deeper (Table S1).

For the rest of the stations (AlpArray and PACASE), one model was prepared per station, with the use of the negative phase depth migrated with IASP91 as the LAB, and 1 vol.% partial melt below it. In case of a few stations (A265A, A272A,





HU04A, HU21A, HU22A, JOS), receiver function data was not sufficient to produce acceptable LAB depth. For most of the

stations, migrated LAB depths are in good agreement (with less than 5 km difference) with the presumed LAB based on the

negative phase depths of Kalmár et al. (2023) (Fig. 7a), with a few outliers (TIH, A266A, HU02A). On this figure, the

distribution of LAB depths with respect to station location and basement depth is reflecting well the different lithospheric

environments within the Pannonian Basin. The majority of the station locations on the Tisza microplate (Fig. 1) is

characterised by thick sedimental coverage (> 1 km) and shallow LAB (50-60 km), representing the areas most affected by

extensional thinning (e.g., Horváth et al., 2006). The stations with minimal to no sediment coverage (< 1 km) have deeper

LAB (70-85 km) are located at or in the vicinity of the Mecsek Mountains (Southern Transdanubia), e.g. KOVH and

MORH.

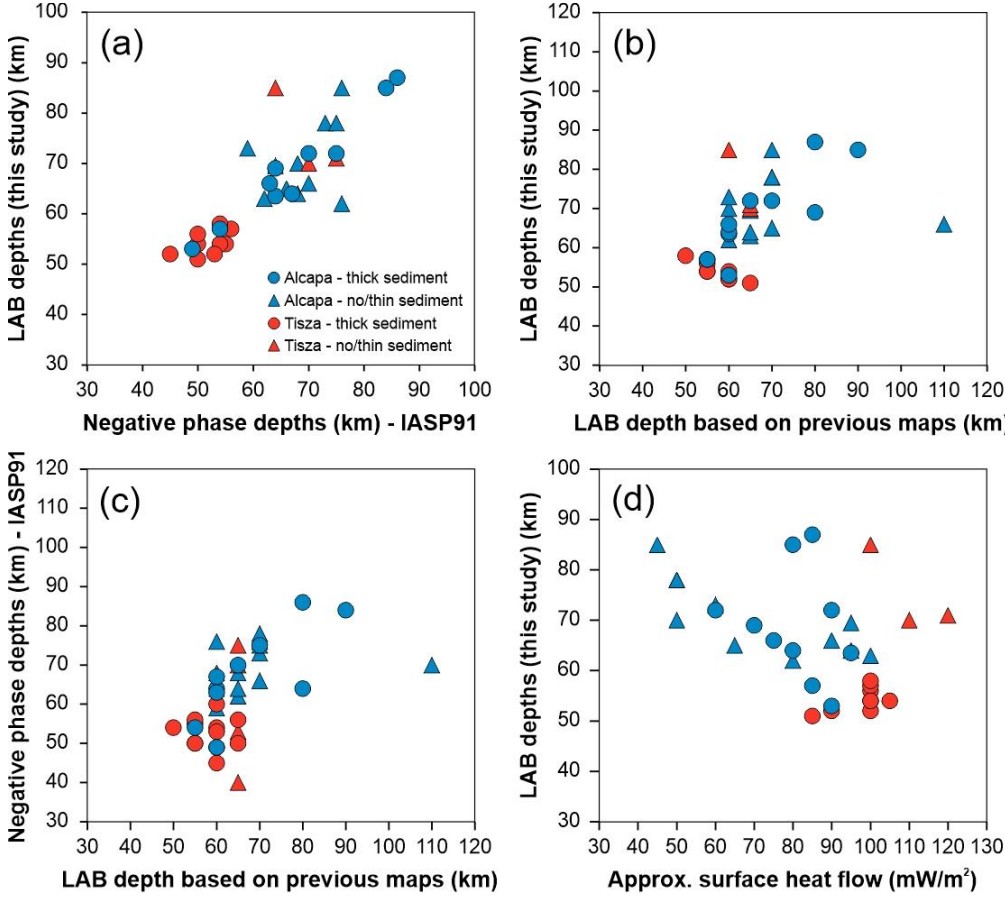

**Figure 7. Comparisons of LAB and negative phase depths obtained with different approaches, as well as surface heat flow for the**
**seismic stations of different location and sediment thickness. a – 1D migrated LAB depths using station-specific forward models**
**considering 1 vol.% melt beneath the LAB vs. negative phase depths of Kalmár et al. (2023). b - 1D migrated LAB depths using**
**station-specific forward models considering 1 vol.% melt beneath the LAB vs. LAB depths based on previous maps (Tari et al.,**
**1999). c – Comparison of previous LAB depths (Tari et al., 1999) and negative phase depths (Kalmár et al., 2023). d –**
**Approximate surface heat flow (Lenkey et al., 2021) vs. 1D migrated LAB depths of this study.**



The Alcapa microplate appears slightly more complex, as stations with different sediment thickness do not show clear distinction. Nevertheless, the Alcapa microplate has stations on various environments. Stations on the northeast part (e.g., ABAH, TRPA) have thick sediment coverage and shallow LAB, similar to the eastern Tisza locations. Stations in the Transdanubian Range, which are characterised by minimal sediment coverage and anomalously low surface heat flow (Lenkey et al., 2021) (e.g., CSKK, MPLH), show LAB depths around 70-75 km. Finally, in the westernmost part of the

Pannonian Basin, stretching towards the foothills of the Alps, the LAB is deeper (~85 km) while sediment coverage is still thick (stations A260A and A262A).

## 6. Discussion

### 6.1. Comparison of LAB depth obtained with local vs. global velocity models

Crustal structure, crust and mantle compositions, temperature, and partial melt in the asthenosphere are all crucial factors in

influencing seismic velocities in the lithosphere and asthenosphere, and thus, the migrated LAB depth. Global velocity models, such as the IASP91 (Kennett and Engdahl, 1991), while provide a good approach for seismic velocities throughout the whole of the Earth's radius, have a rather low resolution on the scale of the lithosphere, which is only a small portion of the profile. The uppermost 120 km of the IASP91 global model, which has previously been used for constraining the LAB depth (or negative phase depth; Kalmár et al., 2023), only has velocity changes at a fixed Conrad and Moho depth, with

constant velocities between them and no account of the LAB. Naturally, the depth of the LAB has a high variance globally, and in areas with thick lithospheres, such as cratons, the LAB is expected deeper than 120 km. Nevertheless, the question arises regarding how accurate 1D migrated LAB depths can be considered for areas with generally shallow LAB and locally variable crustal structures, such as the Pannonian Basin.

On Fig. 8, P- and S-wave velocity profiles are plotted for the four selected stations mentioned earlier (PSZ, EGYH, MORH,

LTVH), based on the models with 1 vol.% partial melt in the asthenosphere, pargasite content increasing to 10 vol% in 5 km above the LAB, and the Moho and LAB (negative phase depths) from Kalmár et al. (2023); i.e., the selected parameter set that was used for 1D migration in all stations of this study. Although the velocities in the lithospheric mantle agree well with that of the IASP91 global model, the patterns of the profiles show several differences (Fig. 8). One of these is the velocity decrease at and below the LAB, caused by the 1 vol.% melt used in our models, a parameter which may be argued (see

section 6.2). The other significant discrepancy is velocity pattern of the crust. Due to the Moho depth being usually shallower than the global model, P- and S-wave velocities increase earlier to reach ~8 and ~4.5 km/s, respectively, which are the values characteristic for the lithospheric mantle. Furthermore, the global model uses constant velocities between discontinuity layers, which also contributes to this difference. On the other hand, our models incorporate the sediment-related velocity drop immediately below the surface. The effect is most prominent in station LTVH, where thick sediment

layer is combined with very shallow Moho (almost coinciding with the Conrad of the global model) and LAB depths.



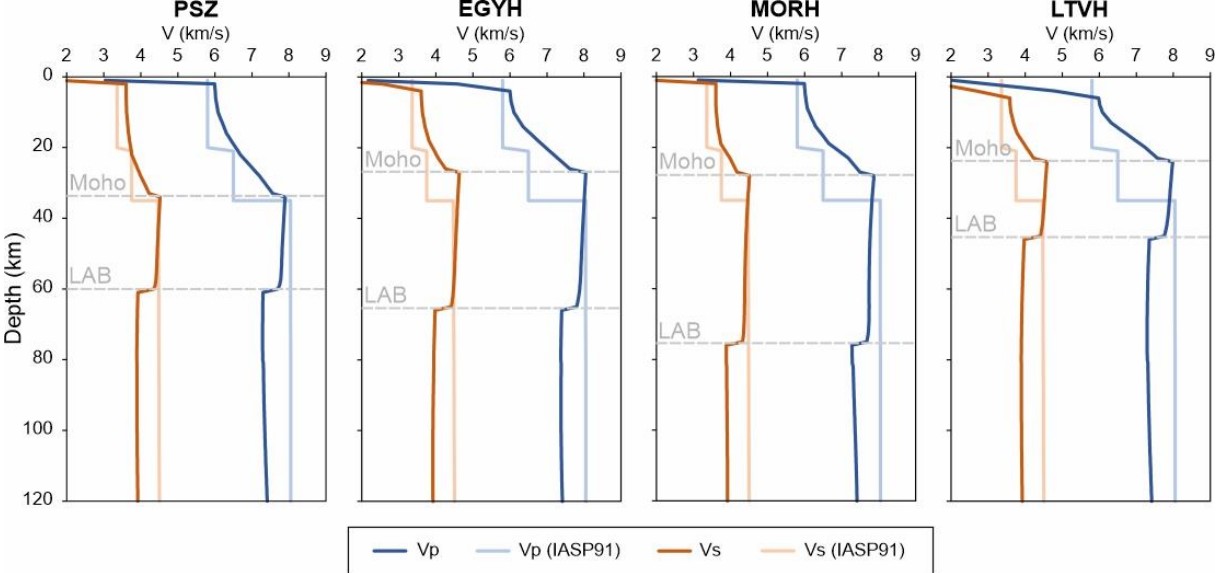

**Figure 8.** Velocity profiles (Vp, Vs) of the four selected stations (PSZ, EGYH, MORH, LTVH) described in section 5.1. The profiles were constructed for each station using the models with 1 vol.% partial melt in the asthenosphere, pargasite content increasing to 10 vol% in 5 km above the LAB, and the Moho and LAB (negative phase depths) from Kalmár et al. (2023). For comparison, the IASP91 global Vp and Vs models are also depicted; the velocity increases at 20 and 35 km corresponding to the Conrad and Moho discontinuities.

The question may arise to what extent these differences affect the migrated LAB depth if the global or a locally customised model is used. LAB depths obtained with these different velocity profiles generally have a strong correlation (Fig. 7a). There are three stations, TIH, A266A and HU02A, which show significant difference in their migrated LAB depth compared to the 1D migrated negative phase depth of Kalmár et al. (2023), who used the modified IASP91 global model. In case of TIH station, weak quality of the stack receiver function (i.e., lack of a coherent negative phase; Figure S3) suggests that the migrated LAB value is not reliable, thus we decided not to use it in further interpretations. Receiver functions of stations A266A and HU02A, however, do not warrant omission. Interestingly, these two stations yielded 85 and 70 km, respectively, by migration using the 3D velocity model (Zhu et al., 2015) by Kalmár et al. (2023), which are significantly closer to the LAB depths acquired with 1D migration using our velocity models. This supports that these LAB depths are more likely realistic, suggesting that our models can be considered more reliable than the IASP91 global velocity model for 1D migration in certain cases, possibly due to local features.

For comparison, we have also plotted the migrated LAB depths of this study, as well as the negative phase depths of Kalmár et al. (2023), against previous data summarised in Tari et al. (1999) (Fig.7 b, c). The data are significantly more scattered, and the correlation is weak (note that the uncertainty is significantly higher, up to ± 10 km for the map of Tari et al., 1999, and the values are approximated due to the interpolation). Kalmár et al. (2023) found that these previous LAB values are more accurate for the central parts of the Pannonian Basin (e.g., the Great Hungarian Plain) than its surrounding areas (Eastern Alps, Carpathians), as previously a quasi-linear increase was assumed for the LAB depth towards the orogenic rims.



Our results confirm this; although discrepancies are only striking for SOP station, where the LAB was previously estimated
to be at 110 km depth, but migration with the global IASP91 and the forward velocity model of this study yielded 70 and 66
km, respectively (Fig. 9a).

## 6.2. Effect of melt beneath the LAB

Although in our forward models we considered partial melt to be present in the asthenosphere, it must be noted that while it
is still unambiguous whether melts are necessary to build such models, there is a growing number of evidence that melt is
present underneath the LAB (i.e., Rychert et al., 2020). The IASP91 global model does not include a change in the seismic
velocities that could be associated with the LAB, yet the seismic low velocity zone  has been known for decades. Several
factors were proposed to be responsible for the velocity decrease besides melt, including elastically accommodated grain
boundary sliding (Karato, 2012), anisotropy (Beghein et al., 2014), change in mantle composition (Selway et al., 2015) or
grain size (Faul and Jackson, 2005), and volatile-bearing minerals (Selway et al., 2015; Kovács et al., 2017, 2021; Saha et
al., 2021). Note that while some of these were proposed in association with the mid-lithospheric discontinuity (MLD)
observed in thick lithospheres, the same depth could correspond to the LAB in areas with thin lithospheres (oceanic and
young continental plates; Kovács et al., 2021). A growing number of studies suggests partial melt or melt layers to account
for the low seismic velocity anomaly (e.g., Thybo and Perchuć, 1997; Thybo, 2006; Rychert et al., 2007; 2020; Kawakatsu et
al., 2009; Naif et al., 2013). As Rychert et al. (2020) explain, melt/fluids is yet the simplest and most straightforward
phenomenon which fits best the velocity behaviour at the LAB/MLD. However, it is still a matter of debate whether melt
could be considered a uniform factor which is globally present in this depth, and its amount is expected to vary on a wide
scale between tectonically active (e.g., mid-ocean ridges, subduction zones) and stable intra-continental (e.g., cratonic)
regions. It has been proposed that the presence of volatiles, due to the breakdown of hydrous phases such as pargasitic
amphibole (Green et al., 2010) and/or the limited capability of water incorporation in nominally anhydrous mantle minerals
(Mierdel et al., 2007) can lead to a lowering of melting temperature (Hirschmann, 2010). Based on these principles, Kovács
et al. (2021) introduced the pargasosphere hypothesis, suggesting that the LAB in thin-lithosphere areas is marked by the
pargasite dehydration solidus, and the MLD in thick continental lithospheres is caused by sub-solidus pargasite breakdown
and the consequential liberation of water-rich fluids.

Our models showed that while small amount of partial melt below the LAB has a significant effect on seismic velocities
(Fig. 5), this effect is not that pronounced on the final result of the 1D migration. As mentioned earlier, above a certain
presumed LAB depth, the change in the seismic velocity does not affect the migrated LAB. Furthermore, the melt curves
flatten out towards shallower presumed LAB values (Fig. 6d, f), which suggests that here the migrated LAB depth majorly
depends on the melt content, and not the presumed LAB depth. There is a range of ~10 km, depending on the station, where
the migrated LAB values quickly increase until they reach the 'maximum'. It is noteworthy that the migrated LAB tends to
be at greater depth compared to the presumed LAB, when considered smaller (and likely more realistic) melt amounts. This
may be a consequence of how the models are constructed and the model velocities are calculated.



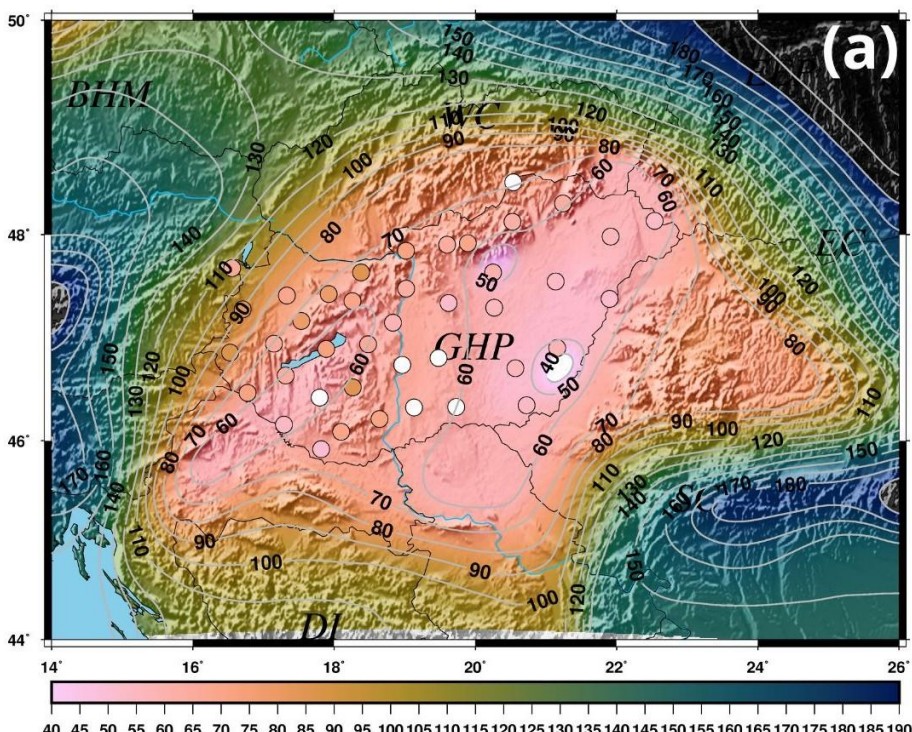

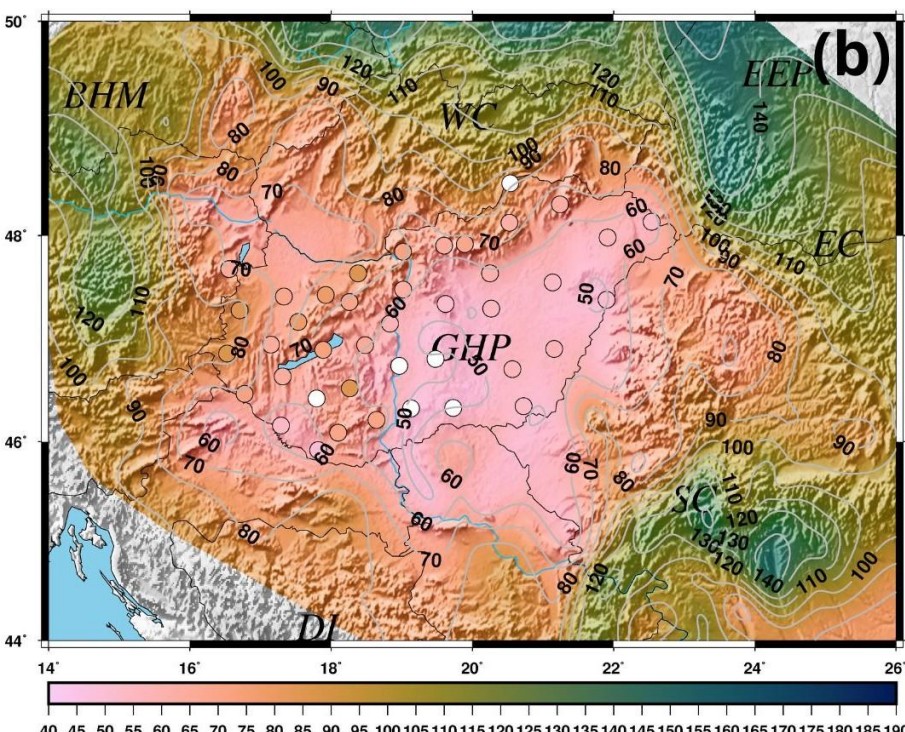



**Figure 9. 1D migrated LAB depths of the 41 seismological stations superimposed on (a) the previous lithospheric thickness map of Tari et al. (1999) and (b) the 1D negative phase depth map of Kalmár et al. (2023). Black areas on (a) are off the scale. Stations**
**depicted in white had insufficient receiver function data to obtain a coherent LAB depth. The maps were prepared using the Generic Mapping Tools software (Wessel et al., 2019) with colour scale from the Scientific Colour Maps by Crameri et al. (2020). Abbreviations are as on Fig. 1.**

## 6.3. Effect of metasomatism and volatile-bearing phases on seismic velocities

Since modal and/or stealth metasomatism alter the mineral composition of mantle bodies, they need to be taken into account
when investigating the potential change of seismic velocities in a lithospheric profile. Based on the calculations described in section 4.2, the most extreme difference in seismic velocities would be ~0.8 km/s for P-wave propagation in a purely forsteritic vs. purely diopsidic composition (Fig. 4). Compared to the common mantle lithologies (lherzolite, harzburgite and dunite), pyroxene-rich domains such as websterites or pyroxenites could cause a < 0.5 km/s velocity decrease, however, the significance can be considered negligible unless they are present as large bodies or a coherent layer in the mantle profile.
Based on numerous xenolith data from the Carpathian Pannonian region, the degree of metasomatism can change on small scales (i.e., within a single outcrop) (e.g., Liptai et al., 2017, Aradi et al., 2020; Patkó et al., 2022), and extreme compositions are only present in very small volumes (Szabó et al., 2004). Thus, in general, the effect of metasomatism should not be considered significant in the full extent of a lithologic column.

Another factor to take into account are volatile-bearing minerals, which also lower seismic wave velocities. Pargasitic
amphibole, the most abundant OH-bearing mineral in the mantle, while may be enriched locally in supra-subductional mantle, is generally present in a few vol. % in the uppermost mantle where it is stable up to ~90-100 km (Green et al., 2010, Kovács et al., 2021). It has been proposed by previous studies (e.g., Selway et al., 2015; Saha et al., 2021) that amphibole and other hydrous silicates and carbonates may contribute to the seismic velocity drop at this depth, which result in the global discontinuity equivalent to the LAB in areas with thin lithosphere, and the mid-lithospheric discontinuity in thick
continental lithospheres (Kovács et al., 2021). Yet the question arises regarding the proportion of hydrous phases that could entirely account for the observed velocity drop. Selway et al. (2015) modelled velocities with as high as 25 vol.% amphibole and 1 vol.% phlogopite, and reported > 5% drop in S-wave velocity. Saha et al. (2021) used an assemblage of 10 vol% amphibole, 2.1 vol.% phlogopite and 0.2 vol.% carbonate, which resulted in 2-3% S-wave velocity reduction. Both studies concluded that while such large amounts of hydrous phases can be produced even in the cratonic sub-continental lithospheric
mantle, they are likely not the only factor that play a role in the formation of mid-lithospheric discontinuities (or the LAB). Our results corroborate this, as a gradual amphibole enrichment (up to 10 vol.%) immediately above the LAB only produced ~ 1% velocity decrease, which we consider more realistic, since a global thick layer of hydrous minerals is still a matter of debate. Therefore, we suggest that the distribution of water in the mantle, i.e., the potential enrichment of amphibole or other hydrous phases at depths of 90-100 km alone do not significantly account for the drop in seismic velocities.





**6.4. Implication of surface heat flow**

A question may arise regarding the reliability of surface heat flow in determining temperatures at depth. As Lenkey et al. (2021) explains, anomalous surface heat flow may result from groundwater flow. In case of the Transdanubian Range, groundwater flow of meteoric origin in the karstic carbonate rocks produces a cooling effect, which is responsible for the extremely low (40-60 mW/m$^2$) surface heat flow values compared to the rest of the Pannonian Basin. Kovács et al. (2017)

also showed that in the Transdanubian Range, there is a strong decoupling between the depth of the 1100°C isotherm representing the LAB according to the pargasosphere concept, and the previously determined LAB depths (Tari et al., 1999). Further, albeit less drastic negative surface heat flow anomalies (around 80 mW/m$^2$) can be present in areas with thick sedimentary layers, such as the northwest and southeast of Hungary, because the sediments have not reached thermal equilibrium yet due to fast sedimentation rate (Békési et al., 2018; Lenkey et al., 2021).

In general, a negative correlation can be expected between surface heat flow and LAB depth. When plotting the stations used in this study (Fig. 7d), however, the data looks more scattered with two apparent trends. One goes from ~85 km LAB depth at ~45 mW/m$^2$ heat flow to ~50 km LAB depth at ~90 mW/m$^2$ heat flow; the other cover about the same LAB depth range but between ~80-110 mW/m$^2$ heat flow. Stations with the extremely low heat flow values, located in the Transdanubian Range or its vicinity, make up about half of the first trend. As mentioned above, these surface heat flow values are likely too low to represent accurate geotherms. Therefore, it is highly likely that stations falling within this trend have their heat flow,

and consequently, temperatures at LAB depth underestimated, whereas the second trend could be considered more realistic. Outliers are KOVH, MORH, and A266A stations, the former two located in the vicinity of a high heat flow area in SW-Hungary, and A266A, as mentioned earlier, is likely to have an LAB depth around 85 km, suggesting that the 100 mW/m$^2$ heat flow may be an overestimation.

**7. Conclusions**

In this study, we employed 1D migration of S-to-P receiver functions, combined with forward modeling of seismic velocity profiles, to refine the depth of the LAB beneath 41 seismological stations across the Pannonian Basin.

By systematically testing key parameters influencing seismic velocities, we demonstrated that partial melt exerts the most significant effect on velocity reduction in the asthenosphere, whereas compositional variations within the lithospheric mantle

and the presence of volatile-bearing minerals such as pargasitic amphibole contribute only minimally. This finding is consistent with recent studies suggesting that even minor amounts of partial melt (<1 vol.%) can significantly alter seismic velocities and account for observed low-velocity anomalies at the LAB.

Our results reveal several critical discrepancies between global velocity models and locally constrained velocity profiles, which impact the accuracy of migrated LAB depths. The most prominent differences arise in the sedimentary and crustal

structures, Moho depth, and the velocity gradient at the LAB. The LAB depths obtained using our local velocity models show strong agreement (within ~5 km) with previously determined negative phase depths derived from IASP91-based

migration. However, in several locations (e.g., A266A and HU02A), LAB depths obtained using local velocity models align better with estimates derived from 3D velocity models, highlighting the necessity of incorporating regional lithospheric complexities into seismic interpretations.

A key finding of our study is the correlation between migrated LAB depths and surface heat flow. Our analysis suggests that anomalously low heat flow values in the Transdanubian Range resulting from crustal groundwater flow cause an overestimation of LAB depths, and thus cannot be considered as representing actual thermal conditions at the LAB. This is evidenced by a subset of stations showing apparent heat flow – LAB depth correlations that deviate from expected trends shown by another group with more reliable surface heat flow measurements.

Our results underscore the importance of using lithology-based forward modeling for constraining seismic velocities in regions with complex lithospheric architecture. While global models such as IASP91 provide a useful reference, they lack the resolution to capture local-scale variations in lithospheric and asthenospheric properties. Integrating location-specific geophysical, petrological, and thermal constraints can contribute to a more accurate interpretation of lithospheric structure.

**Data availability**

The data that support the findings of this study are included in the Supplemetary materials.

**Author contributions**

**NL**: Methodology, Investigation, Visualization, Writing – Original Draft, Writing – Review & Editing

**DK**: Formal Analysis, Software, Visualization, Writing – Review & Editing

**IJK**: Conceptualization, Supervision, Writing – Review & Editing

**Competing interests**

The authors declare that they have no conflict of interest.

**Acknowledgements**

The authors are grateful to all their co-team members within the AlpArray Seismic Network Team (www.alparray.ethz.ch) and the AdriaArray Seismology Group (https://orfeus.readthedocs.io/en/latest/adria_array_main.html) for the temporary
stations, as well as the Central and Eastern European Earthquake Research Network Team (www.ce3rn.eu) that provides the backbone permanent network in the region.





**Financial support**

The reported investigation was financially supported by the National Research, Development and Innovation Fund (Grant PD142953) and NN141956 TopoTransylvania project, as well as an MTA FI Lendület Pannon LitH$_2$Oscope Research grant (LP2018-5/2019-2023).

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
