# Peer review of "Constraining the depth of the lithosphere-asthenosphere boundary in tectonically complex regions using locally adjusted lithological forward models and seismic velocities"

_EGUsphere, 2025_

## Referee Comment (RC1)

Constraining the depth of the lithosphere-asthenosphere boundary in tectonically complex regions using locally adjusted lithological forward models and seismic velocities

by N. Liptai, D. Kalmár and I. J. Kovács

**Abstract**

OK

**1 Introduction**

*l.* 36 "lower temperatures" it is better if you write "lower temperatures with respect to the ones expected by the 'thermal' LAB"

*l.* 38 Consider to replace "… these are all subsolidus processes" with "all of these processes occur under subsolidus conditions"

In addition, please define very briefly what is meant by "subsolidus processes", as the term can be ambiguous (e.g., temperature and pressure below the solidus but possibly involving solid-state transformations or deformation).

This part is very clear, I was wondering if you might want to mention some P-to-S studies carried out within the Pannonian Basin, and then explain that your target is deeper, which is why you use S-to-P conversions instead.

*l.* 45-47 Consider to rephrase this sentence for a better clarity, for example:

"Receiver function analysis is a widely used method for determining the depth of seismic discontinuities, as seismic waves convert from shear to compressional (S-to-P) or vice versa (P-to-S) when crossing boundaries between materials with contrasting physical properties".

*l.* 49-50 This statement: "Unlike the sharp Moho interface, the LAB lacks a distinct seismological discontinuity, making its accurate determination challenging" is clear but it would be helpful if the authors could briefly explain why its determination is more challenging (e.g. temperature effects, phase changes, resolution limits, …)

l. 54 Please put "sensu stricto" in italic font

**2. LAB depths in the Pannonian Basin**

- *l.* 79 Please replace "prepared" with "compiled"
- l. 81 To avoid too many repetitions, please replace "map" with "study"
- *l.* 87 You can delete the word "most" at the beginning of the sentence
- l. 90 Replace "used" with "considered"

**3. Methods**

- l. 104 "... and temporary stations..." delete "and"
- *l. 105* just write "Pannonian-Carpathian-Alpine Seismic Experiment PACASE (Schlömer et al., 2024), currently within the framework of the AdriaArray initiative".
- *l.* 116 Replace "For the 1D velocity models, the Excel workbook of Abers and Hacker (2016) was used…" with "For the 1D velocity model construction, we adopt as a reference the work of Abers and Hacker (2016), …"
- *l.* 120-121 "As significant compositional changes are not expected in the mantle of the region, it was used for the models of all stations".

This is probably a strong assumption, please justify this constraint.

- *l.* 125-127 Sorry I do not understand this sentence: "Although crustal structures are considered more complex, this uniform composition represents a sufficiently realistic approximation, and calculated velocities are in good agreement with the results of Kalmár et al. (2021)"
- You say that the crustal structures are considered more complex with respect to? Mantle?
- What do you mean with "good agreement"? Can you give an example of the velocity values computed at some stations with your method and the findings of Kalmár et al. (2021)? You can put one Table or a Figure with some stations here and probably add in the Supp. Material.
- *l.* 136-137 Technical question: why do you use iterative time-domain deconvolution method and not frequency deconvolution?

Do you perform any quality control? Just one sentence here could be very useful.

Probably you can take a look at these works:

- Colavitti, L., & Hetényi, G., & the AlpArray Working Group. (2022). A new approach to construct 3-

D crustal shear-wave velocity models: Method description and application to the Central Alps. Acta

*Geodaetica et Geophysica*, 57(4), 529–562. <a href="https://doi.org/10.1007/s40328-022-00394-4">https://doi.org/10.1007/s40328-022-00394-4</a>.

- Hetényi, G., Plomerová, J., Bianchi, I., Exnerová, H. K., Bokelmann, G., Handy, M., et al. (2018).

From mountain summits to roots: Crustal structure of the Eastern Alps and Bohemian Massif along

longitude 13.3°E. *Tectonophysics*, 744, 239–255. <a href="https://doi.org/10.1016/j.tecto.2018.07.001">https://doi.org/10.1016/j.tecto.2018.07.001</a>.

l. 138 Please write one or two sentences with some characteristics about the 1D migration method

described in Kalmár et al. (2023). This can be very useful for the readers.

4. Parameter tests

4.1 Heat flow

*l.* 147 Please rephrase the sentence in this way: "While pressure can be reliably calculated for a given

depth, estimating the rate of temperature increase (the geotherm) is more challenging, as it can vary

regionally depending on several factors, most notably lithospheric architecture and thermal history"

*l.* 151-153 Probably this sentence "The high heat... of karst systems" needs at least one reference to

support this statement.

*l.* 156 What about the uncertainties of the P- and S-wave velocity profile?

I would not expect to see an error bar on the velocity profile, but it might be worth including a sentence

in the main text addressing the uncertainty associated with the velocity profiles obtained for the

different heat flows.

4.2 Mantle mineral composition

OK

4.3 Pargasite

OK

4.4 Partial melt

OK

**5. Results**

**5.1 Model set on 4 stations**

l. 211 Please replace "building" with "construction"

*l.* 242 "... beyond the negative phase on the receiver function" probably here you can refer to one image of the RFs in the supplement.

**5.2 Models of rest of the stations**

*l.* 246 For which reason, do you separate the model related to the permanent stations and to the temporary ones? Please justify your choice.

Probably you can also think to revise the current title from "Models of rest of the stations" to "Models for the other stations".

l. 256 Replace "On this figure" with "In Fig. 7"

**6. Discussion**

**6.1 Comparison of LAB depth obtained with local vs. global velocity models**

l. 324-326 "Our results... respectively"

consider to rephrase the sentence: "Our results are consistent with this observation. However, significant discrepancies are found only for the SOP station, where the LAB was previously estimated at 110 km depth, while migration using the global IASP91 model and the forward velocity model developed in this study yielded depths of 70 and 66 km, respectively (Fig. 9a)".

**6.2 Effect of melt beneath the LAB**

*l.* 343 "... presence of volatiles" please specify which volatiles you mean.

**6.3 Effect of metasomatism and volatile-bearing phases on seismic velocities**

*l.* 371 Please quantify "very small volumes": this scale is even smaller with respect to the outcrop? *l.* 388 "… is still a matter of debate" please mention some works related to crustal thickness and the content of hydrous minerals.

**6.4 Implication of surface heat flow**

*l.* 391 Please rephrase the opening sentence with something like:

"The reliability of surface heat flow as an indicator of subsurface temperatures may be questioned".

l. 398 Replace "because" with "since"

l. 408 Rephrase like this: "Stations KOVH, MORH and A266A" represent outliers,..."

**7. Conclusions**

*l.* 431-432 At this point: "While global models such as IASP91 provide a useful reference, they lack the resolution to capture local-scale variations in lithospheric and asthenospheric properties" you could consider adding a short remark about the potential use of 3D reference models for a potential future work, which could help capture lateral heterogeneities in the lithosphere and asthenosphere.

I hope my suggestions will be helpful in improving the manuscript,

Best Regards,

Leonardo Colavitti

\_\_\_\_\_

Leonardo Colavitti, PhD

Post Doctoral Research Fellow

University of Genoa (Italy)

DISTAV (Department of Earth, Environment and Life Sciences)

Seismology Lab of RSNI (Regional Seismic Network of Northwestern Italy)

**Figures**

**Figure 1**

Map is fine from my point of view. I suggest just 2 minor comments:

- In the Legend on the top left, you can put the title "Stations" centered with respect to the white rectangle, as you did for the name of the symbol of the stations
- In the inset on the top right, you can color differently the lands (in light yellow) and the seas (light blue)

**Figure 2**

In Fig. c), it is probably better to flip the y-axis, considering the convention you used for Fig. a) and Fig. b) as well.

**Figure 3**

OK

**Figure 4**

OK

**Figure 5**

OK

**Figure 6**

For a better visibility, you can think to put the limit of the y-axis from the panel **(a)**, **(c)**, **(e)** and **(g)** from -0.5 to 4.5 (still using the label from 0 to 4).

**Figure 7**

Unless any particular reason, I think you can limit the vertical axis in **(b)** and **(c)** to the upper limit of 100 km.

**Figure 8**

OK

**Figure 9**

The figure is appropriate as it stands, but I would recommend displaying depth scale values at 10 km intervals instead of 5 km to ensure better readability.

---

## Referee Comment (RC2)

**Review of the manuscript**

**Constraining the depth of the lithosphere-asthenosphere boundary in tectonically complex regions using locally adjusted lithological forward models and seismic velocities**

by Nóra Liptai, Dániel Kalmár, and István János Kovács

The introduction presents a clear aim to apply 1D migration of S-to-P receiver functions within an interdisciplinary framework to improve constraints on lithosphere-asthenosphere boundary depth. However, Section 3 Methods remains partly brief and general, and would benefit from further elaboration.

- The used RF methodology could be described in more detail. While it is understandable that the authors follow the procedure of Kalmar et al. (2023), the details of parameter setting would be good to mention.
- The authors mention the use of data from 41 seismic stations in the region, but provide no information about the data itself (parameters of selection, preprocessing, etc.).
- Also, information about the resulting Receiver functions, such as the number of RFs, back-azimuth
  coverage, and the quality of the RFs, is missing. Only the sums of S-to-P receiver functions are
  presented in Figure S3 (supplement). Further comments on Figure S3 are below.
- Additionally, the migration method would be good to describe slightly in this paper (even though it is already described in Kalmar et al. (2023)).
- At line 105, the right parenthesis is missing.
- At line 106, the AdriaArray initiative is mentioned. Please cite the paper Kolínský et al., 2025.
   AdriaArray a Passive Seismic Experiment to Study Structure, Geodynamics and Geohazards of the Adriatic Plate. Ann. Geophys. 2025, 68 (5), DM555. https://doi.org/10.4401/ag-9284
- At lines 125-127: The statement about the complexity of the structures is vague, there is no clear
  indication of what the authors compare those structures to, and there is no example of what exactly is
  meant by good agreement.

Besides, it is not clear which data were calculated newly (during the work on this paper) and which were already published. In case that some figures have already been published, it should be mentioned (e.g., the stack of LTVH station – compare with Figure 5 in Kalmar et al., 2023).

Regarding Figure S3, it would be good to specify whether the stacks are from all backazimuths or a specific azimuthal sector only, and describe the content more precisely. For readers unfamiliar with the S-to-P receiver functions, it is unclear what is presented.

In Figure 2c, it would be helpful to reverse the y-axis to agree with fig. 2a and 2b.

In Section 5.2, the authors mention that at six stations (A265A, A272A, HU04A, HU21A, HU22A, JOS), receiver-function data were insufficient to produce an acceptable LAB depth. This would be beneficial to include at least as an explanation in Figure S3 (e.g., by showing the accepted peaks).

In Figure 7, it would be helpful to know which symbol belongs to which station, not only to which group (Alcapa or Tisza and thick or no (thin) sediment). Also, keeping the uniform axis scales at subfigures a - d would make the comparison clearer.

References to the seismic networks are missing:

- AlpArray Seismic Network (2015). AlpArray Seismic Network (AASN) temporary component, AlpArray Working Group, doi:10.12686/alparray/z3\_2015
- Hetényi, G., Plomerová, J., Bielik, M., Bokelmann, G., Csicsay, K., Czuba, W., Meier, T., Šroda, P., Wéber, Z., Wesztergom, V., Žlebčíková, H. (2019). Pannonian-Carpathian-Alpine Seismic Experiment [Data set]. International Federation of Digital Seismograph Networks. https://doi.org/10.7914/SN/ZJ\_2019
- Kövesligethy Radó Seismological Observatory (Geodetic And Geophysical Institute, Research Centre For Astronomy And Earth Sciences, Hungarian Academy Of Sciences (MTA CSFK GGI KRSZO)). (1992). Hungarian National Seismological Network [Data set]. GFZ Data Services, doi:10.14470/UH028726

line 466: Babuska → Babuška

Seismic stations are sometimes mentioned as seismic stations, and sometimes as seismological stations (e.g., lines 12, 86, 96, 102, 358, 412). Unification using the phrase "seismic stations" would be good. Similarly, seismological discontinuity (l. 49) and seismological data (l. 50); low velocity zone  $\rightarrow$  low-velocity zone (l. 36, 39, 54, 205, 331).

It would be beneficial to review the text for grammatical accuracy.

I hope the suggestions mentioned above will help to improve the manuscript.

---

## Author Comment (AC1)

**Constraining the depth of the lithosphere-asthenosphere boundary in tectonically complex regions using locally adjusted lithological forward models and seismic velocities**

by N. Liptai, D. Kalmár and I. J. Kovács

Reply to Reviewer 1

Dear Dr. Colavitti,

Thank you very much for your detailed and constructive comments. We have carefully read through your review and modified the manuscript based on your suggestions. We believe that it has significantly improved. Please find below our replies addressing your comments point by point, highlighted in blue italic.

Sincerely,
Nóra Liptai, Dániel Kalmár, István J. Kovács

**Abstract**

OK

**1 Introduction**

*l. 36* "lower temperatures" it is better if you write "lower temperatures with respect to the ones expected by the 'thermal' LAB"

*We modified the text as suggested.*

*l. 38* Consider to replace "… these are all subsolidus processes" with "all of these processes occur under subsolidus conditions"

In addition, please define very briefly what is meant by "subsolidus processes", as the term can be ambiguous (e.g., temperature and pressure below the solidus but possibly involving solid-state transformations or deformation).

*We modified the text as suggested, and added the brief clarification.*

This part is very clear, I was wondering if you might want to mention some P-to-S studies carried out within the Pannonian Basin, and then explain that your target is deeper, which is why you use S-to-P conversions instead.

*We added a sentence referencing the P-to-S and S-to-P studies of Kalmár et al. (2021, 2023), which targeted shallower and deeper discontinuity layers, respectively. However, this part is more of a general description of the application of receiver functions, and the detailing of our goal comes later on in the manuscript.*

*l. 45-47* Consider to rephrase this sentence for a better clarity, for example:

"Receiver function analysis is a widely used method for determining the depth of seismic discontinuities, as seismic waves convert from shear to compressional (S-to-P) or vice versa (P-to-S) when crossing boundaries between materials with contrasting physical properties".

*We rephrased the text for better clarity based on the suggestion.*

*l. 49-50* This statement: "Unlike the sharp Moho interface, the LAB lacks a distinct seismological discontinuity, making its accurate determination challenging" is clear but it would be helpful if the authors could briefly explain why its determination is more challenging (e.g. temperature effects, phase changes, resolution limits, …)

Thank you for this suggestion. We have enhanced the sentence to include the main physical and methodological reasons:
*"Unlike the sharp Moho interface, the LAB lacks a distinct seismological discontinuity, as it primarily reflects a gradual thermal, rheological, and compositional transition rather than a first-order velocity contrast. Temperature-dependent velocity reductions, partial melting, hydration, and anelastic effects tend to produce broad, frequency-dependent seismic gradients, which—together with limited vertical resolution—make the accurate determination of the LAB particularly challenging."*

*l. 54* Please put "sensu stricto" in italic font

*Done.*

**2. LAB depths in the Pannonian Basin**

*l. 79* Please replace "prepared" with "compiled"

*l. 81* To avoid too many repetitions, please replace "map" with "study"

*l. 87* You can delete the word "most" at the beginning of the sentence

*l. 90* Replace "used" with "considered"

*We applied the above suggested small changes.*

**3. Methods**

*l. 104* "… and temporary stations…" delete "and"

*l. 105* just write " Pannonian-Carpathian-Alpine Seismic Experiment - PACASE (Schlömer et al., 2024), currently within the framework of the AdriaArray initiative".

*l. 116* Replace "For the 1D velocity models, the Excel workbook of Abers and Hacker (2016) was used…" with "For the 1D velocity model construction, we adopt as a reference the work of Abers and Hacker (2016), …"

*We applied the above suggested changes.*

*l. 120-121* "As significant compositional changes are not expected in the mantle of the region, it was used for the models of all stations".

This is probably a strong assumption, please justify this constraint.

*We rephrased this part, to clarify that using an average lithospheric composition for the models is justified by the fact that the xenolith compositions do not significantly differ by location.*

*l. 125-127* Sorry I do not understand this sentence: "Although crustal structures are considered more complex, this uniform composition represents a sufficiently realistic approximation, and calculated velocities are in good agreement with the results of Kalmár et al. (2021)"
- You say that the crustal structures are considered more complex with respect to? Mantle?
- What do you mean with "good agreement"? Can you give an example of the velocity values computed at some stations with your method and the findings of Kalmár et al. (2021)? You can put one Table or a Figure with some stations here and probably add in the Supp. Material.

*We modified the sentence, adding that we mean 'more complex' in petrological sense, as opposed to the applied simplified model.*

*By good agreement we mean that the calculated seismic velocities are in a similar range as those provided by Kalmár et al. (2021). For example, S-wave velocities at LTVH station go from ~3.5 km/s at the top of the upper crust to ~4.2 km/s at the bottom of the lower crust (see figs 13 and 15 in Kalmár et al., 2021, and tab 2 of Table S2 in this manuscript. We used this as a confirmation that our calculations for the crustal part of the model are valid, however we do not think this warrants a separate figure as it does not represent a significant finding.*

*l. 136-137* Technical question: why do you use iterative time-domain deconvolution method and not frequency deconvolution?
Do you perform any quality control? Just one sentence here could be very useful.
Probably you can take a look at these works:
- Colavitti, L., & Hetényi, G., & the AlpArray Working Group. (2022). A new approach to construct 3-D crustal shear-wave velocity models: Method description and application to the Central Alps. *Acta Geodaetica et Geophysica*, 57(4), 529–562.
https://doi.org/10.1007/s40328-022-00394-4.
- Hetényi, G., Plomerová, J., Bianchi, I., Exnerová, H. K., Bokelmann, G., Handy, M., et al. (2018). From mountain summits to roots: Crustal structure of the Eastern Alps and Bohemian Massif along longitude 13.3°E. *Tectonophysics*, 744, 239–255.
https://doi.org/10.1016/j.tecto.2018.07.001.

*Thank you very much for this excellent suggestion. We have added a sentence on quality control and included the references you mentioned. We chose the iterative time-domain deconvolution because it is more robust and better suited to handle the large dataset used in this study. Naturally, we had previously tested the differences between the two methods, and the results clearly showed that the iterative deconvolution approach (Kalmár et al., 2021) is much more applicable in the Pannonian Basin than the frequency-domain method (Kalmár et al., 2018).*

*l. 138* Please write one or two sentences with some characteristics about the 1D migration method described in Kalmár et al. (2023). This can be very useful for the readers.

*Based on your suggestion, we enhanced the description of the 1D migration method.*

**4. Parameter tests**

**4.1 Heat flow**

*l. 147* Please rephrase the sentence in this way: *"While pressure can be reliably calculated for a given depth, estimating the rate of temperature increase (the geotherm) is more challenging, as it can vary regionally depending on several factors, most notably lithospheric architecture and thermal history"*

*We rephrased the sentence accordingly.*

*l. 151-153* Probably this sentence "The high heat… of karst systems" needs at least one reference to support this statement.

*We added the reference.*

*l. 156* What about the uncertainties of the P- and S-wave velocity profile?

I would not expect to see an error bar on the velocity profile, but it might be worth including a sentence in the main text addressing the uncertainty associated with the velocity profiles obtained for the different heat flows.

*Your point is doubtlessly valid, yet it is hard to say anything specific about the uncertainties. The velocity profiles are merely calculated from available heat flow and seismic velocity data, and thus the uncertainty is inherited from these data. In fact, the main aim of the parameter tests was to see how much effect using 'wrong' data might cause.*

**4.2 Mantle mineral composition**

OK

**4.3 Pargasite**

OK

**4.4 Partial melt**

OK

**5. Results**

**5.1 Model set on 4 stations**

*l. 211* Please replace "building" with "construction"

*l. 242* "… beyond the negative phase on the receiver function" probably here you can refer to one image of the RFs in the supplement.

*We added the suggested modifications.*

**5.2 Models of rest of the stations**

*l. 246* For which reason, do you separate the model related to the permanent stations and to the temporary ones? Please justify your choice.

*We separated the models related to permanent and temporary stations because the amount and continuity of available data differ significantly between these two station types. Permanent stations provide substantially longer recording periods, allowing us to use a much larger*

*number of high-quality waveforms, which results in more robust and stable receiver function (RF) estimates. In contrast, temporary stations often operate for shorter time intervals; therefore, we only accepted results from those temporary stations where a sufficient quantity and quality of RF data were available. We considered it important to emphasize—both here and as a general methodological point—that a large data volume is a key prerequisite for obtaining reliable, high-quality RF determinations.*

*We added a short explanation regarding this in the manuscript.*

Probably you can also think to revise the current title from "Models of rest of the stations" to "Models for the other stations".

*l. 256* Replace "On this figure" with "In **Fig. 7**"

*We applied the suggested modifications.*

**6. Discussion**

**6.1 Comparison of LAB depth obtained with local vs. global velocity models**

*l. 324-326* "Our results… respectively"

consider to rephrase the sentence: "Our results are consistent with this observation. However, significant discrepancies are found only for the SOP station, where the LAB was previously estimated at 110 km depth, while migration using the global IASP91 model and the forward velocity model developed in this study yielded depths of 70 and 66 km, respectively (Fig. 9a)".

*We modified the sentence.*

**6.2 Effect of melt beneath the LAB**

*l. 343* "… presence of volatiles" please specify which volatiles you mean.

*We specified the volatiles (mainly H2O and C-O-H components).*

**6.3 Effect of metasomatism and volatile-bearing phases on seismic velocities**

*l. 371* Please quantify "very small volumes": this scale is even smaller with respect to the outcrop?

*We slightly modified the sentence to clarify that out of many xenolith samples, only a small number were pyroxene-rich.*

*l. 388* "… is still a matter of debate" please mention some works related to crustal thickness and the content of hydrous minerals.

*We have added reference.*

**6.4 Implication of surface heat flow**

*l. 391* Please rephrase the opening sentence with something like:

"The reliability of surface heat flow as an indicator of subsurface temperatures may be questioned".

*l. 398* Replace "because" with "since"

*l. 408* Rephrase like this: "Stations KOVH, MORH and A266A" represent outliers,…"

*We have modified the section based on the above suggested changes.*

**7. Conclusions**

*l. 431-432* At this point: "While global models such as IASP91 provide a useful reference, they lack the resolution to capture local-scale variations in lithospheric and asthenospheric properties" you could consider adding a short remark about the potential use of 3D reference models for a potential future work, which could help capture lateral heterogeneities in the lithosphere and asthenosphere.

*We added a small enhancement to the last sentence of the Conclusion about using more developed reference models.*

---

## Author Comment (AC2)

**Constraining the depth of the lithosphere-asthenosphere boundary in tectonically complex regions using locally adjusted lithological forward models and seismic velocities**

by N. Liptai, D. Kalmár and I. J. Kovács

Reply to Reviewer 2

Dear Reviewer,

Thank you very much for your detailed and constructive comments. We have modified the manuscript based on your suggestions and we believe that it has significantly improved. Please find below our reply to your comments point by point, highlighted in blue italic.

Sincerely,
Nóra Liptai, Dániel Kalmár, István J. Kovács

**Comments**
The introduction presents a clear aim to apply 1D migration of S-to-P receiver functions within an interdisciplinary framework to improve constraints on lithosphere-asthenosphere boundary depth. However, Section 3 Methods remains partly brief and general, and would benefit from further elaboration.
• The used RF methodology could be described in more detail. While it is understandable that the authors follow the procedure of Kalmar et al. (2023), the details of parameter setting would be good to mention.
• The authors mention the use of data from 41 seismic stations in the region, but provide no information about the data itself (parameters of selection, preprocessing, etc.).
• Also, information about the resulting Receiver functions, such as the number of RFs, back-azimuth coverage, and the quality of the RFs, is missing. Only the sums of S-to-P receiver functions are presented in Figure S3 (supplement). Further comments on Figure S3 are below.
• Additionally, the migration method would be good to describe slightly in this paper (even though it is already described in Kalmar et al. (2023)).

*Thank you very much for this valuable suggestion. Based on your comment, we have rewritten Section 3 to make the workflow, data usage, and methodological choices clearer. We believe that the revised text now addresses all of the raised questions and concerns related to the description. The new version of the paragraph reads as follows:*
*"The S-to-P receiver functions (RFs) used in this study were computed previously by Kalmár et al. (2023), and here we directly used this existing RF dataset for the seismic forward modelling. In this way, information on the LAB depth was obtained from two independent approaches: (1) the seismologically derived RF-based 1D migration and (2) the geochemistry-based forward modelling applied beneath the same stations.*
*The data selection criteria, preprocessing steps, parameter settings (including deconvolution parameters), and the characteristics of the RF dataset (number of RFs, back-azimuthal coverage, and overall quality) are identical to those described in detail in Kalmár et al.*

*(2023) for the same 41 seismic stations, and are not repeated here. A brief summary of the main steps is nevertheless provided for clarity.*

*Raw S-to-P receiver functions were computed using the iterative time-domain deconvolution method (Ligorría & Ammon, 1999) with 300 iterations following Kalmár et al. (2023), and three independent quality control procedures (Hetényi et al., 2018b; Kalmár et al., 2021; Colavitti et al., 2022). The resulting stack S-to-P receiver functions from the all back-azimuthal direction (Figure S3) were then converted from time to depth by a 1D migration method (Kalmár et al., 2023), using a velocity model defined beneath each seismic station. The median value of the negative S-to-P phase (SpN) was accepted as the representative depth for the 1D migration, which was applied at all 41 investigated stations to estimate the negative phase depth (interpreted as the presence of the LAB) within the lithosphere. For the migration, the velocity models were implemented at a vertical resolution of 5 km, down to a maximum depth of 120 km.''*

• At line 105, the right parenthesis is missing.

*Thank you, we corrected.*

• At line 106, the AdriaArray initiative is mentioned. Please cite the paper Kolínský et al., 2025. AdriaArray – a Passive Seismic Experiment to Study Structure, Geodynamics and Geohazards of the Adriatic Plate. Ann. Geophys. 2025, 68 (5), DM555. https://doi.org/10.4401/ag-9284

*We have added this reference.*

• At lines 125-127: The statement about the complexity of the structures is vague, there is no clear indication of what the authors compare those structures to, and there is no example of what exactly is meant by good agreement.

*We modified the sentence, adding that we mean 'more complex' in petrological sense, as opposed to the applied simplified model.*

*By good agreement we mean that the calculated seismic velocities are in a similar range as those provided by Kalmár et al. (2021). For example, S-wave velocities at LTVH station go from ~3.5 km/s at the top of the upper crust to ~4.2 km/s at the bottom of the lower crust (see figs 13 and 15 in Kalmár et al., 2021, and tab 2 of Table S2 in this manuscript. We used this as a confirmation that our calculations for the crustal part of the model are valid, however we do not think this warrants a separate figure as it does not represent a significant finding.*

Besides, it is not clear which data were calculated newly (during the work on this paper) and which were already published. In case that some figures have already been published, it should be mentioned (e.g., the stack of LTVH station – compare with Figure 5 in Kalmar et al., 2023). Regarding Figure S3, it would be good to specify whether the stacks are from all back azimuths or a specific azimuthal sector only, and describe the content more precisely. For readers unfamiliar with the S-to-P receiver functions, it is unclear what is presented.

*Thank you for your suggestions, we enhanced the caption of Fig. S3 and added clarifications in the main text (the last part of Section 3), see our reply to the first group of comments.*

In Figure 2c, it would be helpful to reverse the y-axis to agree with fig. 2a and 2b. In Section 5.2, the authors mention that at six stations (A265A, A272A, HU04A, HU21A, HU22A,

JOS), receiver-function data were insufficient to produce an acceptable LAB depth. This would be beneficial to include at least as an explanation in Figure S3 (e.g., by showing the accepted peaks).

*Thank you very much for this observation. All values related to the LAB depth are presented in Table S1, while in Figure S3 the stacked SRFs of the stations clearly show, in a visual way, the reasons why certain results were not accepted. For the sake of clarity, we added the following sentence to the manuscript:*
*". In this case, neither the positive nor the negative peak appeared on the SRF, because the data quality was poor or the station operated only for a short period and therefore could not record a sufficient amount of data."*

In Figure 7, it would be helpful to know which symbol belongs to which station, not only to which group (Alcapa or Tisza and thick or no (thin) sediment). Also, keeping the uniform axis scales at subfigures a – d would make the comparison clearer.

*While we agree that it would be useful to know which symbol belongs to which station, due to the large number of data points we think it would be unfeasible to label each station. Nevertheless, all the data presented on this figure are contained in Table S1 for all stations.*

References to the seismic networks are missing:
• AlpArray Seismic Network (2015). AlpArray Seismic Network (AASN) temporary component, AlpArray Working Group, doi:10.12686/alparray/z3_2015
• Hetényi, G., Plomerová, J., Bielik, M., Bokelmann, G., Csicsay, K., Czuba, W., Meier, T., Šroda, P., Wéber, Z., Wesztergom, V., Žlebčíková, H. (2019). Pannonian-Carpathian-Alpine Seismic Experiment [Data set]. International Federation of Digital Seismograph Networks. https://doi.org/10.7914/SN/ZJ_2019
• Kövesligethy Radó Seismological Observatory (Geodetic And Geophysical Institute, Research Centre For Astronomy And Earth Sciences, Hungarian Academy Of Sciences (MTA CSFK GGI KRSZO)). (1992). Hungarian National Seismological Network [Data set]. GFZ Data Services, doi:10.14470/UH028726

*We added these references.*

line 466: Babuska → Babuška

*Thank you, we corrected.*

Seismic stations are sometimes mentioned as seismic stations, and sometimes as seismological stations (e.g., lines 12, 86, 96, 102, 358, 412). Unification using the phrase „seismic stations" would be good. Similarly, seismological discontinuity (l. 49) and seismological data (l. 50); low velocity zone → lowvelocity zone (l. 36, 39, 54, 205, 331).

It would be beneficial to review the text for grammatical accuracy.

*We have unified these expressions that appeared in different versions in the manuscript text.*

I hope the suggestions mentioned above will help to improve the manuscript.